# Integrating 4 methods to evaluate physical function in patients with cancer (In4M): protocol for a prospective cohort study

Gita Thanarajasingam ![ORCID],[1] Paul G Kluetz,[2] Vishal Bhatnagar,[2] Abbie Brown,[3] Elizabeth Cathcart-Rake,[4] Matthew Diamond,[2] Louis Faust ![ORCID],[5] Mallorie H Fiero,[2] Scott F Huntington,[6,7] Molly Moore Jeffery ![ORCID],[8,9] Lee Jones,[10] Brie N Noble,[11] Jonas Paludo,[1] Brad Powers,[12] Joseph S Ross,[6,13,14] Jessica D Ritchie,[13] Kathryn J Ruddy,[4] Sarah E Schellhorn,[6,7] Michelle E Tarver,[2] Amylou C Dueck,[11] Cary P Gross[6,7,13,15]

**Correspondence to**
Dr Gita Thanarajasingam;
thanarajasingam.gita@mayo.edu

## ABSTRACT

**Introduction** Accurate, patient-centred evaluation of physical function in patients with cancer can provide important information on the functional impacts experienced by patients both from the disease and its treatment. Increasingly, digital health technology is facilitating and providing new ways to measure symptoms and function. There is a need to characterise the longitudinal measurement characteristics of physical function assessments, including clinician-reported outcome, patient-reported ported outcome (PRO), performance outcome tests and wearable data, to inform regulatory and clinical decision-making in cancer clinical trials and oncology practice.

**Methods and analysis** In this prospective study, we are enrolling 200 English-speaking and/or Spanish-speaking patients with breast cancer or lymphoma seen at Mayo Clinic or Yale University who will receive intravenous cytotoxic chemotherapy. Physical function assessments will be obtained longitudinally using multiple assessment modalities. Participants will be followed for 9 months using a patient-centred health data aggregating platform that consolidates study questionnaires, electronic health record data, and activity and sleep data from a wearable sensor. Data analysis will focus on understanding variability, sensitivity and meaningful changes across the included physical function assessments and evaluating their relationship to key clinical outcomes. Additionally, the feasibility of multimodal physical function data collection in real-world patients with breast cancer or lymphoma will be assessed, as will patient impressions of the usability and acceptability of the wearable sensor, data aggregation platform and PROs.

**Ethics and dissemination** This study has received approval from IRBs at Mayo Clinic, Yale University and the US Food and Drug Administration. Results will be made available to participants, funders, the research community and the public.

**Trial registration number** NCT05214144; Pre-results.

## STRENGTHS AND LIMITATIONS OF THIS STUDY

⇒ This study addresses an important unmet need by characterising the performance characteristics of multiple patient-centred physical function measures in patients with breast cancer or lymphoma. Physical function is an important and undermeasured clinical outcome. Scientifically rigorous capture and measurement of physical function constitutes a key component of cancer treatment tolerability assessment both from a regulatory and clinical perspective.

⇒ This study will include patients with lymphoma or breast cancer receiving a broad range of cytotoxic chemotherapy regimens. While recruitment will occur at two academic sites, patients who ultimately receive treatment at local community sites will be included.

⇒ A patient-centred health data aggregating platform facilitates the delivery of patient-reported outcome measures and collection of wearable data to researchers, while reducing patient burden compared with traditional patient-generated data collection and aggregation methods.

⇒ Heterogeneity in patient willingness or comfort engaging with mobile products including smartphones and wearables, enrolment primarily at large academic centres and the modest sample size are potential limitations to the external validity of the study.

## INTRODUCTION

Cancer clinical trials have long emphasised important metrics of tumour response and survival rates to evaluate the benefit of new cancer treatments. However, there has been increasing recognition of the importance of systematically assessing how patients feel and function— tolerability—while on treatment.[1] Disease-related symptoms, physical function (PF) and toxicity (ie, side effects from

treatment) are core outcomes that have been identified by the US Food and Drug Administration to inform the safety, tolerability and efficacy of an investigational cancer therapy.[2 3]

PF is defined as the ability to carry out day-to-day activities that require physical effort.[4] Symptoms related to a patient's underlying cancer as well as treatment-related toxicity can impact PF. PF can be assessed using multiple complementary approaches. These include clinician or investigator reports (ClinRO; eg, Eastern Cooperative Oncology Group (ECOG) performance status (PS)),[5] patient-reported outcome measures (PROs; eg, questionnaires administered to patients who assess their physical functioning), performance outcome (PerfO) measures[6] involving measurement observation of a patient's function (eg, 6 min walk test (6MWT), Timed Up and Go (TUG) test) and digital health technologies such as wearable sensors. Given that there are multiple approaches to assessing PF, quantitative data are needed to understand differences in measurement characteristics between these distinct data sources.

## Historical approach to evaluating physical function in cancer clinical trials: clinician-reported assessment (ClinRo)

The widely accepted method for recording a patient's overall functional status in most cancer clinical trials has historically been clinician-reported or investigator-reported PS using scales such as the Karnofsky PS[7] and its derivative, the ECOG PS.[5] These tools have become a ubiquitous, international standard in haematology/oncology practice and research. While the simplicity of the PS is attractive, it is also a drawback, as it lacks granularity, which becomes particularly relevant in the setting of patients at ECOG PS 2–3 and clinical trial eligibility. Many trial eligibility criteria exclude patients with ECOG PS>2, thus leaving the subjective judgement of an oncologist as the main factor determinant of whether a patient can receive what is often a highly desirable therapy on study, or not.[8] Additionally, the score is clinician assessed, rather than directly reflecting the patient experience[9] and is rarely assessed post-baseline in most cancer trials.

## Novel and more comprehensive approaches to measuring physical function which complement clinician-reported assessment

### Patient-reported outcomes (PRO)

PROs are reports of the status of the patient's health that come directly from the patient, without interpretation of the patient's response by a clinician or caregiver.[10] PROs are an assessment method that can be used to directly capture many aspects of a patients' health, from individual symptoms to functional domains such as physical, emotional, cognitive and social function, to the broad multidomain concept of health-related quality of life (HRQOL). Clinicians often miss or under-report symptomatic adverse events (AEs) experienced by patients who can lead to physical, psychological and other toxicities going unrecognised.[11 12] The systematic incorporation of PRO assessment to measure symptoms and function that affect patients' HRQOL in cancer clinical trials is now recognised as critical to complement standard tumour, survival, and clinician-reported safety data by patients, clinicians, industry, academics and regulators.[13–15]

Some of the more commonly used PRO measurement systems used in cancer research include the European Organisation for Research and Treatment of Cancer (EORTC) questionnaires,[16] Patient-Reported Outcomes Measurement Information System (PROMIS) questionnaires,[17] and the Functional Assessment of Chronic Illness Therapy (FACIT) questionnaires.[18] Several of these tools include items or subscales that assess physical functioning. The PRO version of the Common Terminology Criteria for Adverse Events (PRO-CTCAE)[19] is a library of important symptomatic AEs that can quantify symptomatic toxicities from the patient perspective and can inform causative symptoms that may impact physical functioning. Additionally, prior studies have demonstrated the benefit of patients (in addition to clinicians) directly reporting their own ECOG PS,[20] and patient-friendly versions of the ECOG PS are available.[21–23]

The Patient Global Impression Scales of Change and Severity (PGI-C/PGI-S) are single item questions used to evaluate the patient's perception of change in PF and severity.[24] These questions are often used to assess meaningful change in PRO scores and other functional measures. There are also questions that are disease specific, and tools designed to focus more specifically on a particular domain such as PF.[25]

### Performance outcomes (PerfO) measures

PerfO measures are defined as a measurement based on standardised task(s) actively undertaken by a patient according to a set of instructions. A PerfO assessment may be administered by an appropriately trained individual or completed by the patient independently.[6] There are a variety of validated PerfO measures that can be used to more objectively measure a patient's physical PS, including the TUG test, the Sit-Rise test, the Short Physical Performance Battery, gait speed and grip strength.[26 27] The TUG has been used to predict falls in a cohort of geriatric patients with cancer, but the others have not been validated in broader cancer cohorts.[28] As these tools are primarily used in geriatric populations, they may not be as discriminating with younger patients who have better baseline physical fitness.

On the other hand, the 6MWT is a comprehensive measure of exercise capacity suitable for a broad age range and has been selected as the PerfO of interest in this study. The 6MWT encompasses components of mobility, endurance and functional capacity.[29–31] It is relatively straightforward to administer, requires little expertise or training for the patient, and involves minimal equipment. The 6MWT has been used in patients undergoing cancer treatment as well as cancer survivors[32 33] and normative values for patients with haematological malignancies have been published.

## Wearable technologies

Wearable products have steadily advanced over the last several years with rapidly evolving sensor technology to measure human movement, such as accelerometers, magnetometers and gyroscopes.[34] Commercially available, consumer-grade wearables capable of tracking movement have become ubiquitous to the general public in recent years.[35] Wearable technology mitigates some of the limitations of self-reported data (eg, avoiding recall bias), and the narrow validity of data generated in tightly controlled research lab environments.[34–36]

Wearables have been used to assess physical rehabilitation of patients with disabilities and elderly or hospitalised patients.[37–40] Both capacity (what a patient can do, such as maximal gait speed) and performance (what a patient does, such as total steps per day) have been measured using wearables when assessing changes in PF.[35] A recent study demonstrated a correlation of heart rate variability measured through a wearable product with PF assessed using the Short Physical Performance Battery scores, TUG scores and self-reported PF (SF-36 physical composite scores).[41] The correlation of average daily steps with the 6MWT, another established capacity assessment, was also reported by a recent study.[42]

Fitbit activity tracking products were selected for this study as they are familiar to consumers and have demonstrated acceptable accuracy for heart rate, step count and moderate to vigorous physical activities when compared with research-grade tracking products.[43–45]

## Unmet needs in the evaluation of physical function in cancer patients

There is an unmet need to better characterise the measurement characteristics of clinician-reported outcome (ClinRo), PRO, PerfO and wearable data to inform selection of measures to meet individual cancer clinical trial objectives. A firm scientific understanding of measurement characteristics including variability, sensitivity and meaningful change across all modalities would advance our ability to make science-driven trial design decisions and best inform regulatory and clinical decision-making. Operational aspects including ease of use and adherence are also critical to identify methods to reduce missing data.

Few studies have demonstrated the logistical feasibility, sensitivity and complementarity of different PF measurement modalities in the cancer treatment context. There has been no clear identification of meaningful levels of change for these measures either with respect to patient experience or in correlation with AE or hospitalisation rates. Such data would inform potential use of PROs and digital hardware in the design of tolerability endpoints for regulatory review in cancer clinical trials in all phases of medical product (ie, drug, device and biologics) development.

In this prospective study, we will evaluate PF by four assessment modalities in patients with breast cancer or lymphoma receiving cytotoxic chemotherapy with standard clinical follow-up and care.

## Study aims

The purpose of this study is to integrate four PF methods (ClinRo, PRO, PerfO and wearable data) in a prospective cohort of patients receiving chemotherapy.

There are three main study aims:

1. To measure PF using ClinRo, PRO, PerfO and wearable data: This includes characterising feasibility and assessment challenges by comparing levels of missing data and reasons for missingness across the PF modalities and report on trajectories of function as ascertained by the four PF modalities.
2. To explore associations between various sources of PF data and determine meaningful change thresholds: This includes assessing measurement characteristics of the different modalities, including sensitivity to change and identification of meaningful change thresholds; comparing changes within and between modalities; and exploring associations between changes in the PF modalities and subsequent clinical outcomes, such as patient-reported AEs, other patient-reported domains of HRQOL, acute care usage and chemotherapy dose delay/reduction.
3. To assess patient acceptability and experience using the different PF assessment modalities, via the use of an exit questionnaire, to understand burden and usability of electronic PROs and wearable data collection from the patient perspective.

## METHODS

In this prospective study, we are collecting PF data across the four different assessment modalities in a population of patients with breast cancer or lymphoma receiving routine anticancer therapy including a cytotoxic chemotherapy. We plan to follow patients prospectively for 9 months, tracking clinician and patient self-report of physical functioning, PerfOs and wearable data using a patient-centred health data sharing platform—Hugo Health[46 47]—that will consolidate data from electronic health records (EHRs), patient surveys and wearable data (see figure 1). Patients use their personal smartphone or other web-connected mobile product to answer questionnaires about PF, symptoms and adverse effects. Information from the EHR is collected to record baseline clinical features, ClinRo, treatment plans and outcomes including acute care usage (emergency department visits, hospitalisations) and chemotherapy dose reductions, delays or discontinuations. The focus of this study is to characterise patients' PF trajectories on cancer therapy without any intervention, so no exercise programme or activity guidance are given.

The study is based at Mayo Clinic (Minnesota) and Yale University. Participants are recruited both at community and academic hospitals, as well as clinics affiliated with these sites. Participants can be treated after recruitment at

**Figure 1** In4M study schema. AE, adverse event; ECOG-PS, Eastern Cooperative Oncology Group-performance status; HRQOL, health-related quality of life; PRO, patient-reported outcome.

a local community site and followed remotely after study consent and enrolment is obtained at the primary site. Informational flyers are placed in waiting rooms of breast cancer and lymphoma clinic practices at both primary sites. Charts of potential study candidates are reviewed by clinical investigators, and if potentially eligible, patients are approached about and consented for the study by the study research assistants. Each site will enrol 100 patients. Complete inclusion and exclusion criteria are in online supplemental appendix 1.

### Measures and data collection

A detailed description of Hugo Health, the electronic health data aggregating technology used to administer PRO questionnaires, collect patient EHR portal data and aggregate wearable data in this study, has been published previously.[46 47] All of the data and records described below and generated during this study are kept confidential in accordance with institutional policies and the Health Insurance Portability and Accountability Act on subject privacy.

### ClinRo and PerfO

ClinRo is recorded from the medical record into a REDcap form by research assistants every 3 months. The 6MWT is performed once at baseline (prior to start of chemotherapy) and at 3 months for participants treated at Mayo Clinic and Yale primary sites. Changes in performance between the two time points may be a result of learning effects rather than true change in performance, which is a potential limitation. Participants receiving care at a site other than Mayo Clinic Rochester or Yale University sites will not have an additional 6MWT observation.

### Patient-reported outcomes

Questionnaires are sent by Hugo to patients throughout the 9-month follow-up period (online supplemental table 1). PROs assessing PF include the PROMIS V.2.0 PF 8c short form, PF questions from the EORTC QLQ-F17 instrument, a patient-adapted version of the ECOG PS (PRO-ECOG), and the PGI-C/PGI-S items pertaining to PF. Additional PROs that capture global assessments of

QOL and well-being (functional and QOL domains of the EORTC QLQ-F17 and selected items from the PRO-CTCAE, FACIT GP5) are used to assess the correlation of PF data with symptomatic toxicities, patient-reported AEs, and other domains of HRQOL. Hugo sends automated reminders if patients do not complete the weekly survey after 48 hours or the monthly survey after 1 week. Additionally, at key time points, research assistants call patients if questionnaires have not been completed after 5 days for weekly questionnaires or after 2 weeks and 2 days for the monthly questionnaires.

### Wearable data

A Fitbit model with built-in GPS, the Fitbit Inspire, is used in this study. Multiple data parameters are recorded from the lead-in time point to the completion of month 9 of follow-up. The lead-in time, for baseline data collection prior to initiation of cancer-directed therapy, was pragmatically derived to be at least 24 hours. Fitbit data are automatically uploaded from the wearable to Fitbit's servers when the Bluetooth feature on the patient's wearable is turned on. Hugo downloads that data through the Fitbit API regularly and links it to the other participant data.

Patients are instructed to (1) wear the Fitbit as much as possible during the day and night, limiting non-wear time to recharging periods (approximately 1–2 hours every 3 days) and (2) synchronise (upload) the Fitbit data from the wearable to Fitbit's servers every 3 days using the Fitbit smartphone application. Reminders to synchronise Fitbit data are delivered by Hugo to study participants on a weekly basis.

Predefined parameters evaluating both capacity and performance measurements of PF from three domains (steps/distance, heart rate and activity level) will be used for comparison with the other PF assessment modalities. Additional metrics of interest derived from the raw data parameters or obtained directly from Fitbit will be considered. These additional metrics may include distance walked per day, sleep duration per day, heart rate variability, sleep cycle duration, etc. Reporting non-adherence and

abandonment will include a visualisation of participant drop-out over time accompanied by the total number of participants who dropped out and a distribution of time in the study. Among those who remained in the study, we will report the total remaining, number of days deemed compliant, as well as weeks considered compliant as defined by our completeness criteria. Participants who would no longer like to contribute their wearable data, but are interested in continuing to complete the PROs, are able to stay enrolled in the study.

## Exit questionnaire

An exit questionnaire designed specifically for this study is administered to all participants at month 9 to assess patients' perceptions of their own PF, their feedback on surveys completed during the study, and their perspective on the wearable device. A full copy of the exit questionnaire is provided in online supplemental appendix 2.

## Analysis plan

Specific aim 1: In order to characterise assessment challenges, completion rates will be computed and reasons for missing data will be described. For each PF metric, the completion rate will be computed at applicable time points using (1) a fixed denominator method using all patients ever enrolled and (2) a variable denominator method using the number of active patients at each time point. For the variable denominator approach, at each time point, active participants are those who have not died and have not withdrawn from study participation. Intercurrent events including reason for study withdrawal, disease progression and death will be summarised in analysis.

To describe distributions of PF responses over time, the trajectory of each PF metric will be graphically explored using stream (spaghetti) plots and mean plots. Mean plots will employ raw means as well as estimated means from a general linear mixed modelling at each time point. Estimation will include group means and group mean changes from baseline.

Specific aim 2: To identify measurement characteristics of each PF metric, standard psychometric analyses investigating sensitivity to change and meaningful change thresholds will be carried out. These analyses will employ both anchor-based and distribution-based methods. The primary anchor will be PGI-C and the key secondary anchor will be PGI-S.

Distribution-based analyses for each PF metric will include the mean, SD, median, first quartile, third quartile, minimum and maximum. Effect sizes representing small, moderate and large effects will be computed as 0.2, 0.5 and 0.8 times the baseline SD.[48]

Anchor-based analyses will estimate the mean change for each PF metric over time according to how patients respond to the PGI-C and PGI-S items. Mean change at each postbaseline time point will be described using the mean and SD within strata of patients grouped by their status change (those reporting worsening status, no

change in status and improved status) and their current limitations in PF (no limitations, mild or moderate limitations and severe limitations). Additionally, the standardised response mean will be computed as the mean change score divided by the SD of the change scores within each change category (worsening vs no change vs improvement) or severity category (normal vs mild/moderate vs severe). Values greater than 0.8 will be considered large and values between 0.5 and 0.8 will be considered moderate.[48] Additionally, Spearman correlations between the change in each PF metric and the change in other anchors (eg, physician-reported and patient-reported ECOG PS, patient-reported role function, global health status/QOL and HRQOL via the EORTC QLQ-C17; PRO-CTCAE symptomatic AE grades and FACIT GP5) will be computed. Correlations values of 0.1, 0.3 and 0.5 will be interpreted as small, moderate and large.[48]

The relationship between change in PF metrics and PGI-C and PGI-S items will be investigated using general linear mixed models. Mean change from baseline with 95% CIs will be computed for each PF metric based on mixed modelling. Mixed models will include all PF metrics as outcomes and time as a categorical variable. Additional patient or design characteristics will be incorporated as baseline covariates. Composite covariance will initially be used, with the final covariance structure selected based on minimisation of the Akaike information criterion. All patients who consent for participation in this study and complete at least one PF metric will be included in statistical analysis. In the primary analysis, all observations available will be used.

We will conduct secondary analyses, assessing the association between baseline patient characteristics and baseline PF metrics, using Spearman correlations and longitudinal PF metrics using statistical modelling. Key baseline patient characteristics that will be explored as feasible based on the distribution of the characteristics observed in the sample will include cancer cohort (breast vs lymphoma); age (<65 vs ≥65 years); physician-reported ECOG PS; patient-reported ECOG PS; patient-reported role function, global health status/QOL and HRQOL via the EORTC QLQ-C17; PRO-CTCAE symptomatic AE grades; and FACIT GP5. Association between longitudinal patient characteristics (patient-reported ECOG PS; patient-reported role function, global health status/QOL and HRQOL via the EORTC QLQ-C17; PRO-CTCAE symptomatic AE grades; and FACIT GP5) and longitudinal PF metrics will be explored using Spearman correlations at successive time points as well as statistical modelling (bivariate linear mixed modelling).

Specific aim 3: Statistical analysis will be primarily descriptive for the exit questionnaire data. Free-text responses will be coded for themes by two independent reviewers. Continuous responses in the exit survey will be summarised using means, SD, medians, minimums and maximums. Categorical responses including adjudicated

themes from free-text responses will be summarised using frequencies and relative frequencies.

Power considerations: Our targeted sample enrolment is 200 patients, which we expect will allow the team to have data available for a given PF metric at early post-baseline time points (at least the first 3 months) for at least 170 patients. Based on a prior study evaluating association between PF as measured by the QLQ-C17 and a PGI-C item assessing physical condition,[19] we anticipate 25% of patients to report worsening and the mean change in PF among these patients to be −8.2 points. The remaining 75% of patients reporting no change or improvement had a mean change in PF of 0.9 points (pooled SD 15.0). Thus, with a sample size of 170 patients, this study has 92% power to detect a similar change as the prior study using a t-test comparison with a two-sided alpha of 0.05. Statistical analysis will employ a modelling approach across all time points and thus power estimation based on a single time point can be considered conservative.

Missing data: Missing data from patient questionnaires will be handled in a number of ways. Missing items within a summary or scale score will be handled according to each questionnaire's published scoring algorithms. When summary or scale score data are missing, baseline patient/disease characteristics will be compared between patients who do and do not provide data for a given analysis and patterns of missing data will be graphically explored. All analyses will first be completed using all available data, then by integrating missing categories for categorical data and analyses completed using multiple imputation via chained equations (20 or more for each analysis), and finally using pattern mixture models for longitudinal analyses. Output from all analyses will be tabulated and descriptively compared with assess the degree to which missing data impacts study results.

For all statistical analyses, p values <0.05 will be considered statistically significant; however, interpretation will take into consideration that type I error is not strictly controlled across all planned analyses. For interpreting the clinical significance of effects, 0.2, 0.5 and 0.8 SD effects will be considered as small, moderate and large.

### Data collection and management

The Hugo platform will aggregate data from the EHR, PROs and wearables. At study enrolment, patients provide Hugo access to their health portals by authenticating themselves using their username and password. PerfO and clinician-reported ECOG will be among data collected by the research assistant and entered into a secure REDCap database. Additionally, clinical coinvestigators will review the medical records of each patient directly for more granular information on tolerability parameters, such as reasons for hospitalisations or dose reductions, and these data are entered into the study REDCap database by the research assistant.

### Patient and public involvement

Three patient-advocate coinvestigators provided input on the design of the study, the selection of PRO survey items, and timing of scheduled assessments and the burden on patients. They also cocreated a 'study welcome letter' to describe in patient-tailored language the purpose of the study, and they have participated in the writing and review of this manuscript. Patient advocates were not involved in the conduct of the study.

### Study limitations

Although patients on this study can receive their breast cancer or lymphoma treatment at primary or local sites as part of this clinical study, recruitment is limited to patients seen at least once at Mayo Clinic or Yale clinical sites, limiting participation to patients who have the physical and financial ability to access these tertiary cancer care centres. Most participants receive treatment at the primary sites and may not be representative of a larger community oncology practice. We do not offer patients a smartphone or other web-connected product if they do not have one, which may limit participation, though smartphone adoption is high at 85% of American adults, including a majority of those with low income and those living in rural areas, with minimal gaps by race and ethnicity.[49] Some patients who already use a non-Fitbit wearable product or are apprehensive of wearable data collection may decline participation. Lastly, we do not have formalised technology support for patients over and above the research assistants in this study, which may limit our ability to swiftly address technical issues related to Hugo or Fitbit.

## ETHICS AND DISSEMINATION

Institutional review board (IRB) approval was secured at Mayo Clinic, Yale University and the US Food and Drug Administration. Any protocol modifications will be submitted for IRB approval prior to implementation, and all trial registration details will be updated accordingly. Study results will be disseminated through publications in general, and specialty medical journals and conferences.

### Study update

At the time of this publication, all sites have obtained local IRB approval and are enrolling participants. The COVID-19 pandemic delayed study activation at both sites; enrolment in this study began in January 2022. A total of 146 participants have been enrolled at the time of this manuscript submission.

**Author affiliations**
[1]Division of Hematology, Mayo Clinic, Rochester, Minnesota, USA
[2]US Food and Drug Administration, Silver Spring, Maryland, USA
[3]Health Education and Content Services, Mayo Clinic, Rochester, Minnesota, USA
[4]Department of Medical Oncology, Mayo Clinic, Rochester, Minnesota, USA
[5]Division of Health Care Delivery Research, Kern Center for the Science of Health Care Delivery, Mayo Clinic, Rochester, Minnesota, USA
[6]Department of Internal Medicine, Yale School of Medicine, New Haven, Connecticut, USA

[7]Yale's Cancer Outcomes, Public Policy, and Effectiveness Research (COPPER) Center, Yale School of Medicine, New Haven, Connecticut, USA
[8]Division of Health Care Delivery Research and Department of Emergency Medicine, Mayo Clinic, Rochester, Minnesota, USA
[9]Department of Emergency Medicine, Mayo Clinic, Rochester, Minnesota, USA
[10]Patient Advocate, Arlington, Virginia, USA
[11]Department of Quantitative Health Sciences, Mayo Clinic, Phoenix, Arizona, USA
[12]CancerHacker Lab, Boston, Massachusetts, USA
[13]Yale-New Haven Center for Outcomes Research and Evaluation (CORE), Yale School of Medicine, New Haven, Connecticut, USA
[14]Department of Health Policy and Management, Yale School of Public Health, New Haven, Connecticut, USA
[15]Department of Chronic Disease Epidemiology, Yale School of Public Health, New Haven, Connecticut, USA

**Correction notice** This article has been corrected since it was published. Missing middle initials have been added in the author names.

**Contributors** PGK, GT, CPG, VB, MMJ, MD, MET, JSR, JDR, AB, LJ, BP, ACD and KJR were involved in conception or design of the work. MD, LF, BNN, JDR, MHF and ACD planned for acquisition, analysis or interpretation of the data. GT, EC-R, SFH, JP, KJR, SES and CPG were involved in screening, enrolment and health record review. GT and CPG drafted the work. All authors critically revised the work for important intellectual content and provided final approval of the version to be published. GT and CPG agree to be accountable for all aspects of the work in ensuring that questions related to the accuracy or integrity of any part of the work are appropriately investigated and resolved.

**Funding** This work was supported by the Food and Drug Administration (FDA) of the US Department of Health and Human Services (HHS) as part of a financial assistance award (U01FD005938) totalling US$2 665 476 with 100% funded by FDA)/HHS.

**Disclaimer** The contents are those of the author(s) and do not necessarily represent the official views of, nor an endorsement, by FDA/HHS, or the U.S. Government.

**Competing interests** GT has received research funding from the from the Food and Drug Administration for the Yale-Mayo Clinic Center of Excellence in Regulatory Science and Innovation (CERSI) (U01FD005938) that directly supports this work. She also receives grant funding from the US National Cancer Institute (NCI) U01 Tolerability Consortium (U01CA 2330463), and the Mayo Clinic Center for Clinical and Translational Research (CTSA) (KL2TR 023794). CPG has received research funding from the NCCN Foundation (Astra-Zeneca) and Genentech, as well as funding from Johnson and Johnson to help devise and implement new approaches to sharing clinical trial data. Over the past three years, MMJ reports grant funding from the US Food and Drug Administration, National Institutes on Drug Abuse, Centers for Disease Control and Prevention, Agency for Healthcare Research and Quality, American Cancer Society, and the National Center for Advancing Translational Sciences. JSR currently receives research support through Yale University from Johnson and Johnson to develop methods of clinical trial data sharing, from the Food and Drug Administration for the Yale-Mayo Clinic Center of Excellence in Regulatory Science and Innovation (CERSI) (U01FD005938), from the Medical Devices Innovation Consortium as part of the National Evaluation System for Health Technology (NEST), from the Agency for Healthcare Research and Quality (R01HS022882), from the National Heart, Lung and Blood Institute of the National Institutes of Health (NIH) (R01HS025164, R01HL144644), and from Arnold Ventures for the Collaboration for Regulatory Rigor, Integrity, and Transparency (CRRIT); in addition, Ross is an expert witness at the request of Relator's attorneys, the Greene Law Firm, in a qui tam suit alleging violations of the False Claims Act and Anti-Kickback Statute against Biogen. JDR currently receives research support through Yale University from Johnson & Johnson to develop methods of clinical trial data sharing and from the US Food and Drug Administration for the Yale-Mayo Clinic Center of Excellence in Regulatory Science and Innovation (CERSI) (U01FD005938). SFH has received consulting fees outside of this work from Janssen, Genentech, AbbVie, Flatiron Health, BeiGene, AstraZeneca, ADC Therapeutics, Epizyme, Merck, Seattle Genetics, TG Therapeutics, Tyme, Pharmacyclics, SeaGen, and Arvinas. Over the past three years, KJR reports grant funding from the US Food and Drug Administration, National Institutes of Health, United States Department of Defense, and American Cancer Society. SES has received consulting fees from Eisai, Celgene, SeaGen, and Cardinal Health. She has previously received research funding to her institution from Genetech and Pfizer. All other authors have no relevant conflicts of interest to disclose.

**Patient and public involvement** Patients and/or the public were involved in the design, or conduct, or reporting, or dissemination plans of this research. Refer to the Methods section for further details.

**Patient consent for publication** Not applicable.

**Provenance and peer review** Not commissioned; externally peer reviewed.

**ORCID iDs**
Gita Thanarajasingam http://orcid.org/0000-0003-2144-5415
Louis Faust http://orcid.org/0000-0002-9741-7894
Molly Moore Jeffery http://orcid.org/0000-0003-3854-6810

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
