## [Reviewer comments · BMJ Open]

ARTICLE DETAILS

TITLE (PROVISIONAL)	Integrating 4 Methods to Evaluate Physical Function in Patients with Cancer (In4M): Protocol for a prospective cohort study
AUTHORS	Thanarajasingam, Gita; Kluetz, Paul; Bhatnagar, Vishal; Brown, Abbie; Cathcart-Rake, Elizabeth; Diamond, Matthew; Faust, Louis; Fiero, Mallorie H.; Huntington, Scott; Jeffery, Molly; Jones, Lee; Noble, Brie; Paludo, Jonas; Powers, Brad; Ross, Joseph; Ritchie, Jessica; Ruddy, Kathryn; Schellhorn, Sarah; Tarver, Michelle; Dueck, Amylou; Gross, Cary

VERSION 1 – REVIEW

REVIEWER	Singh, Favil Edith Cowan University
REVIEW RETURNED	11-May-2023

GENERAL COMMENTS	Minor comments Throughout the manuscript and including the title - the use of patient with cancer is correct but best to be specific to breast cancer and lymphoma patients. In the methods under Clinro and Perfo, another site was suggested - please elaborate on this. Similarly in this same sentence, how will not having a 6MWT at this other side effect or impact the comparison in terms of patient recruitment? it seems that the authors have already collected data on this and this protocol looks retrospective rather than prospective. under PRO - there seem to be so many PRO questions- how would the authors consider or prevent patient questionnaire fatigue? This is further exacerbated by weekly questions for the first 2 months. why not just keep it on a monthly basis rather than weekly for the first 2 months? Again, I understand that you have now collected this data for more than half of your sample size. Under wearable data - Were there any instructions to the patients about planned exercise or physical activity - if there were what was it? If not, please elaborate on this in the manuscript. Please elaborate on the exit questionnaire. 6MWT - why not measure at the end of the study at M9 too - what was the reason not to do this and only at 2-time points?
---

REVIEWER	Edbrooke, Lara The University of Melbourne, Physiotherapy, School of Health Sciences
REVIEW RETURNED	05-Jun-2023

GENERAL COMMENTS

Thanks for the opportunity to review this manuscript titled 'Integrating 4 Measures to Evaluate Physical Function in Patients with Cancer (In4M): Protocol for a prospective study' for publication in BMJ Open. The manuscript outlines the protocol for a study currently recruiting to characterise longitudinal measurement characteristics (variability, sensitivity and meaningful changes) of a range of different physical function tests in people with breast cancer or lymphoma receiving chemotherapy. The study will also report on physical function relationships to clinical outcomes and feasibility of data collection. The manuscript will make an important contribution to scientific knowledge on this topic and I have made only a few suggestions below for the authors to consider in a revised version of the manuscript.

Title – rather than 4 'measures' perhaps the title would be more accurate if it was 4 'approaches' to measure... - given for the PRO approach several measures are being used

Introduction

Page 11 lines 32-41 – would be more informative if the strength of the correlations mentioned in this section were presented (e.g., r values or 'fair', 'good' etc as defined by Cohen)

Brief background regarding the development of 'Hugo Health' would help the reader to understand how the measurements are truly 'patient-centred'. For example were patients or carers involved in its design? Readers can then refer to the previous publications for additional information.

Methods

PerfO

Page 15 line 24 – the 6MWT is being performed once, yet the guidelines from the ATS/ERS state that 2 tests should be performed to account for any potential learning effects (Holland et al, ERJ, 2014). Changes in performance which are measured between the two timepoints could be a result of learning effects rather than true change in performance and this should be stated as a limitation to the study methods.

Wearable data

Wear of devices for 9 months day and night, with data uploads every 3 days, is asking a lot of participants, particularly when most clinical trials have a 7-10 day wear period at each assessment timepoint and include data so long as 4 days of 8 hours data/day is available.

Given data collection is continuous over the 9 months how will missing data be reported (e.g., will it be clear if participants have met minimum data requirements as outlined above)?

Will participants be able to withdraw from this component of the study and continue completing the other assessments if they wish to?

Exit questionnaire

Is the questionnaire based on a framework (e.g., The Theoretical Framework of Acceptability) or a previously developed questionnaire? Does it include open-ended responses and if so how will these be analysed?

	Appendix 1 Inclusion criteria point 10 is missing a word – ‘...able to regularly upload data from the Fitbit to a XXX in a way that...’ Figure ‘Enrollment’ has an * but there is no legend to denote what this means Table 1 – -all abbreviations in the table must be provided in full in the footnote and abbreviations in the footnote (e.g., CRA) also need to be provided in full for first use. -It is not clear what the final column labelled ‘**’ means
--	--

VERSION 1 – AUTHOR RESPONSE

Reviewer: 1

Throughout the manuscript and including the title - the use of patients with cancer is correct but best to be specific to breast cancer and lymphoma patients.

Response: We appreciate this suggestion. We have reviewed the manuscript again and made the appropriate changes to terminology where appropriate and not redundant.

In the methods under Clinro and Perfo, another site was suggested - please elaborate on this. Similarly in this same sentence, how will not having a 6MWT at this other site effect or impact the comparison in terms of patient recruitment? It seems that the authors have already collected data on this and this protocol looks retrospective rather than prospective.

Response: Thank you for bringing this to our attention. For clarity, we updated the third sentence to “Participants receiving care at a site other than Mayo Clinic Rochester or Yale University sites will not have an additional 6MWT observation.”

In the In4M study, participants are recruited at Yale, community sites affiliated with Yale, and Mayo Clinic Rochester, and can be treated at a local community site and followed remotely after study consent and enrollment is obtained at the primary Yale or Mayo site (as per the 2nd paragraph of the Methods section). Participants who receive treatment at a site outside of Yale University, a community site affiliated with Yale, or Mayo Clinic Rochester have a 6-minute walk test at baseline, but not in follow up. While it would have been ideal to measure the 6MWT longitudinally in patients at all sites, there are pragmatic challenges in arranging for this to be done at centers remote to our own where our study staff are not available. Currently, only a minority of patients are treated outside of Mayo Rochester and Yale University sites, so we do not anticipate this will substantially impact any comparisons of data.

Under PRO - there seem to be so many PRO questions- how would the authors consider or prevent patient questionnaire fatigue? This is further exacerbated by weekly questions for the first 2 months. Why not just keep it on a monthly basis rather than weekly for the first 2 months? Again, I understand that you have now collected this data for more than half of your sample size.

Response: As opposed to an interventional clinical trial, when PRO assessments are integrated with other requirements that can place burden on patients such as research blood draws, research imaging and research clinic visits, in the In4M study, the assessments are limited to the PROs and the two 6-minute walk tests (in addition to passively collected wearable data) only. Patient advocates were involved in the design of the study and approved the number of PRO questions at each time

point. Participants are specifically consented with the PRO assessment schedule in mind, and additionally they are compensated for their time in filling out questionnaires. The questionnaires intentionally vary in length to ask only the questions that were felt to be necessary at any given timepoint, and most of the weekly questionnaires are shorter. FDA Guidance includes a schedule of assessments that includes weekly PROs even in large phase 3 trials, to characterize adverse events and impacts on function that can be worse in the first couple of months on a new cancer therapy. Additionally, with the Hugo platform, participants are able to complete surveys at home, in office waiting rooms, or anywhere that is convenient, at a time that is convenient to them. Of the patients enrolled on the study thus far, the completion rates are excellent which suggests burden is manageable. However, understanding survey fatigue is an important component of the study and is assessed on the exit survey (now included in Appendix 2, question 6).

Under wearable data - were there any instructions to the patients about planned exercise or physical activity - if there were, what was it? If not, please elaborate on this in the manuscript.

Response: There was no specific instruction to patients about planned exercise or physical activity. The focus of the study was to characterize patients' physical function as it is on cancer therapy, not to provide an exercise program or modify the physical activity. We have added the following sentence to the first paragraph of the methods: "The focus of this non-interventional study is to characterize patients' physical function trajectories on cancer therapy without any intervention, so no exercise program or activity guidance are given."

Please elaborate on the exit questionnaire.

Response: The exit questionnaire was designed specifically for this study, to assess patients' perceptions of their own physical function, the surveys completed during the study, and the wearable device. A full copy of the exit questionnaire is now provided in Appendix 2. A paragraph clarifying this has also been added to the Methods section.

6MWT - why not measure at the end of the study at M9 too? What was the reason not to do this and only at 2-time points?

Response: We agree with the reviewer that it would have been ideal to measure the 6MWT at the end of study, as well as at other time points throughout treatment. Nonetheless, there are pragmatic clinical challenges in arranging for this to be done. When patients come for a treatment day, they have lab tests drawn, see their care providers, and then typically proceed directly to the chemotherapy unit for treatment on the same day, with little break in between. It was felt it would be burdensome to require participants to meet with the study coordinators post chemotherapy, and most patients are not physically present other than on the day of treatment to complete this in-person assessment.

Reviewer: 2

Thanks for the opportunity to review this manuscript titled 'Integrating 4 Measures to Evaluate Physical Function in Patients with Cancer (In4M): Protocol for a prospective study' for publication in BMJ Open. The manuscript outlines the protocol for a study currently recruiting to characterise longitudinal measurement characteristics (variability, sensitivity and meaningful changes) of a range of different physical function tests in people with breast cancer or lymphoma receiving chemotherapy. The study will also report on physical function relationships to clinical outcomes and feasibility of data collection. The manuscript will make an important contribution to scientific knowledge on this topic

and I have made only a few suggestions below for the authors to consider in a revised version of the manuscript.

Title – rather than 4 ‘measures’ perhaps the title would be more accurate if it was 4 ‘approaches’ to measure... - given for the PRO approach several measures are being used

Response: The reviewer’s point is well taken. We chose the title in part for this abbreviation which suggests we are “informing” the study of physical function in cancer. However, we appreciate the suggestion and will change “measures” to “methods” in the title of the study.

Introduction

Page 11 lines 32-41 – would be more informative if the strength of the correlations mentioned in this section were presented (e.g., r values or ‘fair’, ‘good’ etc as defined by Cohen)

Response: We have added a reference to Cohen’s 1988 book for all effect size statements. We have also added a delineation of small, moderate, and large effect sizes for correlation.

Brief background regarding the development of ‘Hugo Health’ would help the reader to understand how the measurements are truly ‘patient-centred’. For example were patients or carers involved in its design? Readers can then refer to the previous publications for additional information.

Response: Thank you for your comment. The data aggregation platform, Hugo Health, is considered ‘patient-centered’ due to its ability to consolidate data from ‘patient-centered’ sources, such as electronic health records and patient surveys. Patients can complete surveys on their mobile devices or computer on their own time which is generally felt more patient-centered than in-clinic paper surveys. Additionally, by gathering electronic health record data to identify information on parameters such as hospitalizations or emergency visits, patients do not have to be contacted directly, which thus reduces participant burden. We have included the website reference to the Hugo Health platform for further details (reference #47, Dhruva SS et al, NPH Digit Med 2020;3:60.).

Methods

PerfO

Page 15 line 24 – the 6MWT is being performed once, yet the guidelines from the ATS/ERS state that 2 tests should be performed to account for any potential learning effects (Holland et al, ERJ, 2014). Changes in performance which are measured between the two timepoints could be a result of learning effects rather than true change in performance and this should be stated as a limitation to the study methods.

Response: The 6MWT is being performed twice, once at baseline and again at 3 months (page 13) in the majority of patients. In a small minority of patients who are not treated at Mayo Clinic Rochester, Yale University or Yale affiliated community sites, we are unable to perform the 2nd assessment of the 6MWT. At current status, this is a very small group of patients on the study and we anticipate most participants will have 2 assessments for robust comparison. The reviewer’s point about learning effects rather than true change in performance is a valid limitation. We have included this in the Methods, paragraph 4.

Wearable data

Wear of devices for 9 months day and night, with data uploads every 3 days, is asking a lot of participants, particularly when most clinical trials have a 7-10 day wear period at each assessment timepoint and include data so long as 4 days of 8 hours data/day is available.

Given data collection is continuous over the 9 months how will missing data be reported (e.g., will it be clear if participants have met minimum data requirements as outlined above)?

Response: We agree that we set a high bar for data completeness at the outset of the study due to the fact that this study will inform regulatory decision making. That being said, we will report all data available including data that met our criteria for completeness (valid week defined as 4 days with at least 2 weekend days, 10 hours a day of wear time) as well as data received that did not meet these criteria.

Reporting non-adherence and abandonment will include a visualization of participant drop out over time accompanied by the total number of participants who dropped out and a distribution of time in the study. Among those who remained in the study, we will report the total remaining, number of days deemed compliant, as well as weeks considered compliant as defined by our completeness criteria. This information has now been added to the text, in the Methods section, Wearable data subsection, 2nd to last paragraph.

Will participants be able to withdraw from this component of the study and continue completing the other assessments if they wish to?

Response: Yes, participants who would no longer like to contribute their wearable data but are interested in continuing to complete the PROs are able to stay enrolled in the study. This has now been clarified in the Methods, end of section on "Wearable data."

Exit questionnaire

Is the questionnaire based on a framework (e.g., The Theoretical Framework of Acceptability) or a previously developed questionnaire? Does it include open-ended responses and if so how will these be analysed?

Response: The exit questionnaire was designed specifically for the In4M study and is now included in Appendix 2 of this paper. It does include open-ended responses, which will be coded for themes by two independent reviewers. Adjudicated themes will be statistically analyzed using descriptive statistics. This information has been clarified in the text of the manuscript.

Appendix 1

Inclusion criteria point 10 is missing a word – '...able to regularly upload data from the Fitbit to a XXX in a way that...'

Response: Thank you for this comment- this has been updated.

Figure

'Enrollment' has an * but there is no legend to denote what this means

Response: Thank you for mentioning this- the * has been removed.

Table 1 –

-all abbreviations in the table must be provided in full in the footnote and abbreviations in the footnote (e.g., CRA) also need to be provided in full for first use.

Response: Thank you for this comment- the table has been updated.

-It is not clear what the final column labelled '***' means

Response: Thank you for mentioning this. We removed the column with "***" from the table.

VERSION 2 – REVIEW

REVIEWER	Edbrooke, Lara The University of Melbourne, Physiotherapy, School of Health Sciences
REVIEW RETURNED	03-Aug-2023
GENERAL COMMENTS	Thank you for submitting a revised version of the manuscript and addressing the questions I raised in my initial review.